# Video-CT MAE: Self-supervised Video-CT Domain Adaptation for Vertebral Fracture Diagnosis

**Lukas Buess**[1,3]                                              Lukas.Buess@tum.de

**Marijn F. Stollenga**[4]                                    stollenga@imfusion.com

**David Schinz**[2]                                              david.schinz@tum.de

**Benedikt Wiestler**[2]                                          b.wiestler@tum.de

**Jan S. Kirschke**[2]                                          jan.kirschke@tum.de

**Andreas Maier**[3]                                          andreas.maier@fau.de

**Nassir Navab**[1]                                              Nassir.Navab@tum.de

**Matthias Keicher**[1]                                    Matthias.Keicher@tum.de

[1] *Computer Aided Medical Procedures, Technical University of Munich, Germany*

[2] *Klinikum Rechts der Isar, Technical University of Munich, Germany*

[3] *Pattern Recognition Lab, Friedrich-Alexander-Universität Erlangen-Nürnberg, Germany*

[4] *ImFusion GmbH, Munich, Germany*

**Editors:** Accepted for publication at MIDL 2024

## Abstract

Early and accurate diagnosis of vertebral body anomalies is crucial for effectively treating spinal disorders, but the manual interpretation of CT scans can be time-consuming and error-prone. While deep learning has shown promise in automating vertebral fracture detection, improving the interpretability of existing methods is crucial for building trust and ensuring reliable clinical application. Vision Transformers (ViTs) offer inherent interpretability through attention visualizations but are limited in their application to 3D medical images due to reliance on 2D image pretraining. To address this challenge, we propose a novel approach combining the benefits of transfer learning from video-pretrained models and domain adaptation with self-supervised pretraining on a task-specific but unlabeled dataset. Compared to naïve transfer learning from Video MAE, our method shows improved downstream task performance by 8.3 in F1 and a training speedup of factor 2. This closes the gap between videos and medical images, allowing a ViT to learn relevant anatomical features while adapting to the task domain. We demonstrate that our framework enables ViTs to effectively detect vertebral fractures in a low data regime, outperforming CNN-based state-of-the-art methods while providing inherent interpretability. Our task adaptation approach and dataset not only improve the performance of our proposed method but also enhance existing self-supervised pretraining approaches, highlighting the benefits of task-specific self-supervised pretraining for domain adaptation. The code is publicly available at https://github.com/lbuess/Video-CT_MAE.

**Keywords:** Vertebral Fracture Diagnosis, Domain Adaptation, Self-supervised Learning

## 1. Introduction

Spinal health is a critical aspect of overall well-being and quality of life. Early and accurate diagnosis of vertebral body anomalies is essential for appropriately treating spinal disorders. Osteoporotic fractures, for instance, affect up to 12% of men and women aged 50-79 years across Europe (Harvey et al., 2010). CT has become an indispensable tool for diagnosing

vertebral fractures. However, manual interpretation of CT scans can be time-consuming and subjective, potentially leading to errors and delays in diagnosis and treatment (Carberry et al., 2013). Deep learning has already demonstrated promising results in automating the detection of vertebral fractures (Husseini et al., 2020; Engstler et al., 2022; Keicher et al., 2023). However, these studies have also highlighted the need for interpretable methods, as understanding the decision-making process of the models is crucial for building trust and ensuring reliable clinical application. Vision Transformers (ViTs) (Dosovitskiy et al., 2020) have shown promise for this due to their inherent interpretability through attention visualizations. However, their application has been primarily limited to 2D medical images (Chład and Ogiela, 2023), as they are data-hungry and often rely on initialization from models pretrained on large-scale 2D image datasets like ImageNet (Deng et al., 2009). This limits their effectiveness in tasks involving volumetric data, such as vertebral fracture detection in CT images. A potential solution is using models pretrained on videos, which are also 3D data with spatial and temporal dimensions (Ke et al., 2023). Video-pretrained models offer a promising solution for initializing ViTs for 3D medical image analysis, but the domain shift between videos and medical images is substantial. An alternative approach is to use self-supervised pretraining of ViTs with in-domain data, which has been shown to improve anatomical image understanding and enhance downstream task performance (Tang et al., 2022). Surprisingly, we find that published self-supervised ViT pretraining models significantly underperform on our task compared to CNN models, which show similar performance whether randomly initialized or pretrained on CT patches using Models Genesis (Zhou et al., 2021). While there are many publicly available CT datasets containing spine images, vertebral fractures are rare, and datasets including these annotations are few, with a high imbalance of classes. We argue that pretraining on a task-specific unlabeled dataset with self-supervised methods, even though it contains mainly healthy vertebrae, can help the model to understand the anatomy and improve performance in detecting pathologies. Therefore, we curate a task-specific pretraining dataset and propose a novel approach that combines the benefits of transfer learning and self-supervised pretraining for vertebral fracture detection in CT scans. Our main contributions are:

- We propose a framework that allows Vision Transformers to effectively detect vertebral fractures in 3D CT images despite a low data regime, outperforming CNN-based methods while providing inherent interpretability through attention visualizations.

- We introduce a self-supervised domain adaptation method and a new task-specific pretraining dataset to bridge the gap between video-pretrained models and medical images, enabling the learning of relevant anatomical features in the target domain.

- Our thorough experimental evaluation demonstrates the effectiveness of the proposed task-specific pretraining in improving downstream task performance for both existing pretraining methods and our novel adaptation of video-based transfer learning.

## 2. Related Work

**Vertebral Fracture Detection**    Deep learning-based vertebral body classification methods can be divided into 2D and 3D approaches. 2D methods analyze a single sagittal slice

from the 3D vertebra volume (Husseini et al., 2020), which is efficient and allows initialization with ImageNet weights but fails to capture important 3D structural features. More advanced 2.5D approaches aggregate information from multiple 2D slices using RNNs (Bar et al., 2017) or LSTM networks (Tomita et al., 2018) but still do not fully exploit the 3D spatial context of CT data. Recent studies predominantly utilize 3D models to leverage the comprehensive volume data (Engstler et al., 2022; Keicher et al., 2023; Chettrit et al., 2020), with the initial application of 3D convolutions in vertebral fracture detection pioneered by Nicolaes et al. (2020). However, these models cannot use pretrained ImageNet weights and must be trained from scratch, which is challenging given the limited labeled medical datasets.

Recent studies are concentrating more on applying transformer-based techniques to classify vertebrae. Chład and Ogiela (2023) explored the effectiveness of transformers in identifying cervical spine fractures from single 2D slices. Similarly, Windsor et al. (2022) employed a hybrid approach, combining a CNN for 2D feature extraction from sagittal slices with transformers for feature aggregation within and across scans. However, these methods have limitations in capturing 3D structural features due to their reliance on 2D inputs.

**Self-supervised Learning**  In recent years self-supervised learning has emerged as a popular approach for pretraining deep learning models. A recent trend in computer vision is transformer-based masked image modeling approaches, which have been inspired by the success of masked language modeling in NLP as demonstrated by BERT (Devlin et al., 2018). Masked image modeling has already been well-established in computer vision for 2D images (He et al., 2022), videos (Feichtenhofer et al., 2022) and multimodal models (Girdhar et al., 2023). More recent methods like MSN (Assran et al., 2022) and I-JEPA (Assran et al., 2023) improve efficiency by using joint-embedding architectures, avoiding pixel reconstruction unlike traditional approaches.

Most famous self-supervised learning methods were initially designed for natural images, but there have been significant advancements in the medical field as well. One noteworthy contribution is Models Genesis, an approach that involves pretraining a CNN-based model (Zhou et al., 2021). Two other prominent methods, both rooted in transformer-based techniques, leverage the fusion of multiple pretext tasks such as image restoration, contrastive learning, and image rotation prediction (Tang et al., 2022).

Current self-supervised pretraining methods face two key challenges: Using non-medical data offers less domain-specificity for tasks like CT-based vertebral fracture detection, while medical-specific self-supervised learning is limited by smaller datasets compared to the natural image and video domains.

**Video Pretraining for CT Analysis**  Addressing the issue of deep learning's reliance on large labeled datasets like ImageNet (Deng et al., 2009) in medical applications, recent research has underscored the benefits of leveraging extensive video datasets for pretraining 3D medical models (Zunair et al., 2021). Ke et al. (2023) and Rajpurkar et al. (2020) both found that pretraining 3D medical models on large-scale, out-of-domain video datasets yields better performance than training from scratch or using conventional in-domain CT datasets. These models, pretrained in a different domain, encounter drawbacks when applied to medical CT data without appropriate domain adaptation.

## 3. Method

Our study involved creating a dataset comprising 27,776 unlabeled vertebra crops (refer to section Experimental Setup 4 for more details). However, it's noteworthy that our dataset's volume is still small when compared to extensive datasets like ImageNet (Deng et al., 2009) and Kinetics-700 (Carreira et al., 2019). Transformer-based models typically necessitate training on expansive datasets (Li et al., 2023), prompting our approach to leverage transfer learning already in the pretraining stage. Specifically, we explore using pretrained weights from data-rich domains to boost self-supervised pretraining for our vertebra CT data. Initializing CT models with ImageNet weights appears suboptimal, as they lack the capacity to capture crucial 3D details present in CT volumes but absent in 2D images. Therefore, we are focusing on using weights pretrained in the video domain. Videos, embodying 3D spatiotemporal data, present a closer alignment with the characteristics of CT volumes, potentially offering a more effective foundation for our research.

Our video-CT domain adaptation method comprises three steps (see Figure 1). First, we employ weights from self-supervised pretraining on the Kinetics-700 video dataset, using video MAE pretraining (Feichtenhofer et al., 2022), to establish our foundation model. The second, domain adaptation step, involves adapting these weights for vertebra CT data by pretraining on an unlabeled vertebra dataset, enhancing model alignment with the CT domain. Finally, we apply these adapted weights to vertebra fracture classification, finetuning the encoder with a labeled dataset.

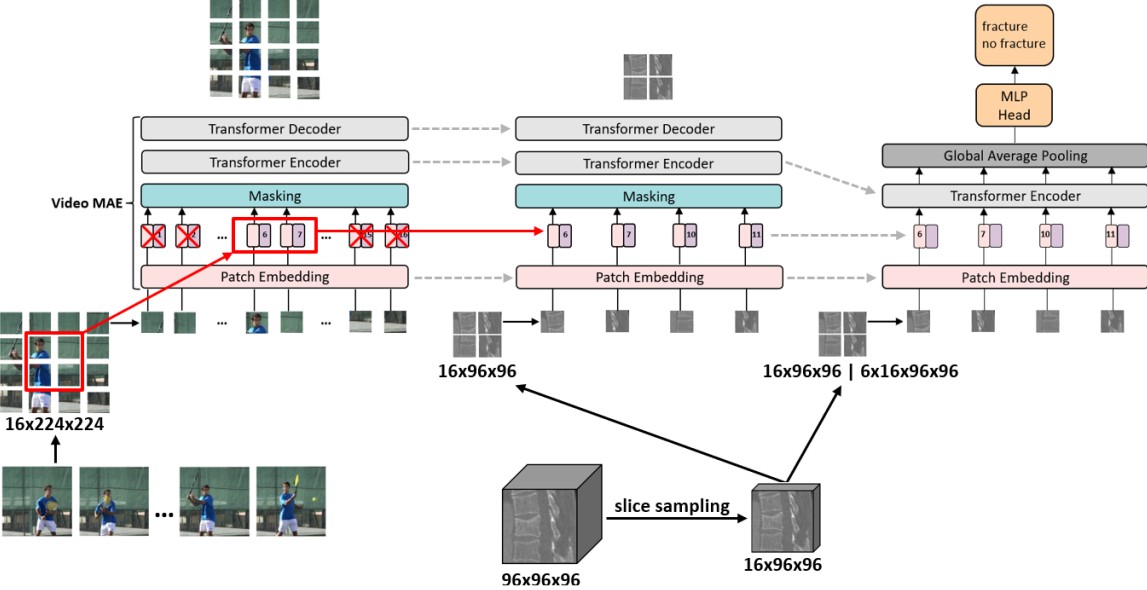

Figure 1: Overview Self-supervised Video-CT Domain Adaptation: 1) video MAE pretraining on the Kinetics-700 dataset 1)-2) positional encoding cropping to initialize domain-specific vertebra CT pretraining 2) domain-specific vertebra MAE pretraining on unlabeled vertebra CT dataset 3) downstream task finetuning

**Video MAE for CT Vertebra Data**   Our approach features two self-supervised pre-training stages, both leveraging the video MAE method (Feichtenhofer et al., 2022). To align the original $96 \times 96 \times 96$ dimensions of vertebra CT volumes with the $16 \times 224 \times 224$ video format from the Kinetics-700 dataset, we select 16 equidistant sagittal slices from the vertebra volume. We use sagittal slices due to their diagnostic relevance in vertebral fracture detection, as they provide crucial information for accurate assessments in this context.

**Positional Encoding Cropping**   After matching the temporal dimension of the video format, the 3D convolutional patch embedding layer ($2 \times 16 \times 16$) from video pretraining can be reused in the vertebra CT pretraining stage. However, with different input sizes ($224 \times 224$ frames for video and $96 \times 96$ slices for CT), input token counts differ (video: $8 \times 14 \times 14$ tokens, CT: $8 \times 6 \times 6$ tokens), which results in a shape mismatch of the positional encodings. To address this, we introduce "positional encoding cropping", preserving only central $96 \times 96$ pixel positional encodings from videos, and discarding outer encodings (Figure 1 red). This allows for the direct initialization of positional encoding weights in domain-specific vertebra CT pretraining using video weights, similar to the approach presented by Kim et al. (2023).

**Slice Sampling**   During vertebra pretraining and downstream task finetuning, we employ different sampling strategies for training and evaluation phases. In the training phase, we sample 16 uniformly distributed sagittal slices. Each training step involves applying a consistent random shift to the chosen slice indices, thereby increasing diversity and effectively serving as an augmentation technique. During the evaluation phase, we maintain our initial sampling strategy but do not shift the selected slice indices. Instead, we use all six possible $16 \times 96 \times 96$ samples covering the entire vertebral volume for prediction. This approach allows aggregation of the prediction by averaging over all sampled subvolumes, ensuring a more robust and representative result. This approach draws inspiration from multi-view testing in the video domain (Feichtenhofer et al., 2019).

## 4. Experimental Setup

**Preprocessing Pipeline**   We developed a preprocessing pipeline designed for vertebra-level classification tasks in CT scans. Initially, a spine segmentation model is applied to identify individual vertebrae. Utilizing the segmentation mask, we then extract $96 \times 96 \times 96$ crops centered on the vertebra of interest.[1] The preprocessing pipeline is illustrated in greater detail in Figure 2 and Figure 3.

**Unlabeled Vertebra Pretraining Dataset**   In our research, we do task-specific self-supervised domain adaptation pretraining by utilizing a big unlabeled vertebra dataset. This dataset was created by collecting seven publicly available CT datasets, each containing spine segments, inspired by the dataset selection in the CTSpine1K dataset (Deng et al., 2021). We subsequently employed the preprocessing pipeline to process these datasets. This approach resulted in a dataset comprising 27,776 individual unlabeled vertebrae extracted from 3,446 different CT volumes. It's essential to note that this preprocessing step relies on a segmentation model, making the process unsupervised. Detailed information about the unlabeled pretraining dataset is summarized in Table 1.

---

1. Segmentation model is provided by the ImFusion GmbH (approach similar to Bürgin et al. (2023))

| Dataset | Patients | Vertebrae |
|---|---|---|
| CT COLONOGRAPHY (Smith et al., 2015) | 784 | 6,515 |
| COVID-19 (An et al., 2020) | 650 | 8,425 |
| MSD-Liver (Simpson et al., 2019) | 201 | 2,297 |
| HNSCC-3DCT-RT (Bejarano et al., 2018) | 31 | 296 |
| DeepLesion (Yan et al., 2018) | 1,107 | 2,820 |
| KiTS21 (Heller et al., 2021) | 300 | 2,727 |
| VerSe (Sekuboyina et al., 2021) | 373 | 4,696 |
| **TOTAL** | **3,446** | **27,776** |

Table 1: Unlabeled Vertebra Pretraining Dataset

**Labeled Vertebra Downstream Task Dataset** To avoid test-leakage between pretraining and downstream classification task finetuning, we strictly separated patients between labeled and unlabeled datasets. For the labeled vertebra dataset we use an in-house dataset from Klinikum Rechts der Isar (Munich) (Foreman et al., 2024) consisting of 6,245 vertebrae (940 of which are fractured) from 457 different patients. More details about the labeled dataset are provided in Table 5.

## 5. Results and Discussion

### 5.1. Ablation Study

This section details our ablation study results, analyzing video transfer learning and self-supervised domain adaptation's impact on the vertebra classification downstream task. Table 2 outlines the performance of the key components in our method.

| Ablated component | F1 (%) | ACC (%) | AUC (%) | AP (%) | FT Min/Ep[2] |
|---|---|---|---|---|---|
| 1) − Video Pretraining | 74.6 | 92.1 | 93.1 | 82.6 | 9 |
| 2) − Vertebra Pretraining | 85.7 | 95.5 | 98.0 | 92.5 | 9 |
| 3) − Multi-View Sampling | 83.7 | 94.6 | 96.5 | 90.1 | 9 |
| 4) − Positional Encoding Cropping | 87.1 | 96.2 | 97.7 | **93.3** | 9 |
| 5) − Vertebra Format Adaptation | 87.5 | 96.0 | 97.5 | 90.7 | 18 |
| 6) − 2) and 4) | 69.1 | 90.9 | 91.8 | 76.5 | 9 |
| **Video-CT MAE** | **88.4** | **96.4** | **98.2** | 93.2 | 9 |

Table 2: Ablation Study: 1) no video pretraining 2) no domain-specific vertebra CT pretraining 3) no multi-view sampling during inference 4) no positional encoding cropping - randomly initialized positional encodings for vertebra CT pretraining 5) original $16 \times 224 \times 224$ video format by adding padding to the $96 \times 96$ slices

**Video Pretraining** The removal of video domain transfer learning significantly reduces performance. This emphasizes the importance of using video domain pretrained weights for our domain-specific vertebra CT pretraining, ensuring a solid foundation and enabling effective transfer of learned features from the video domain to the CT domain.

---

2. Finetuning Minutes/Epoch

**Vertebra Pretraining**  The results demonstrate that video pretraining can be effectively adapted to the CT domain with proper format adjustments, even without domain adaptation. Video pretraining alone outperforms vertebra-only pretraining, aligning with the findings of Ke et al. (2023) and Rajpurkar et al. (2020). However, skipping vertebra pretraining domain adaptation leads to lower performance than the full Video-CT MAE pipeline.

**Multi-View Sampling**  Additionally, multi-view sampling boosts prediction robustness. Combining different views of the vertebra leads to a more reliable final prediction.

**Positional Encoding Cropping**  One can see that by randomly initializing the positional encodings, the performance closely aligns with that of our full Video-CT MAE method. Yet, a closer analysis of the pretraining loss curves offers a significant insight. Employing positional encoding cropping enables a reduction in training epochs (see Figure 4). Another finding is that the direct use of video weights for the downstream task significantly benefits from the positional encoding cropping, as demonstrated in ablation experiment 6.

**Vertebra Format Adaptation**  Using the video model with its original $224 \times 224$ frame size from pretraining yields performance comparable to our full Video-CT MAE method. However, this approach results in increased training time for pretraining and finetuning.

## 5.2. Self-supervised Domain Adaptation

In this section, we conduct a comparative analysis between our Video-CT MAE approach and established self-supervised pretraining methods. Our focus centers on the application of these methods to our vertebra data setup, namely: Models Genesis (Zhou et al., 2021), ViT UNETR (Tang et al., 2022), Swin UNETR (Tang et al., 2022), and MAE (He et al., 2022). We study the importance of task-specific pretraining by: First, random initialization for downstream task finetuning; second, finetuning with publicly available weights; and finally, task-specific self-supervised pretraining using the public weights for initialization.

| Method | Pretraining Data | F1 (%) | ACC (%) | AUC (%) | AP (%) |
|---|---|---|---|---|---|
| Models Genesis 3D | - | 85.8 | 95.6 | 97.8 | 92.1 |
| Models Genesis 3D | 623 CT images[3] (public) | 85.9 | 95.7 | **98.1** | 92.3 |
| **Models Genesis 3D** | **public → vertebrae** | **87.1** | **96.1** | 97.8 | **92.9** |
| ViT UNETR | - | 34.2 | 56.4 | 65.2 | 22.2 |
| ViT UNETR | 771 CT images[4] (public) | 55.9 | 86.8 | 83.7 | 64.3 |
| **ViT UNETR** | **public → vertebrae** | **73.6** | **91.6** | **94.0** | **84.1** |
| Swin UNETR | - | 36.2 | 61.7 | 69.6 | 26.9 |
| Swin UNETR | 5.050 CT images[5] (public) | 57.0 | 86.9 | 83.3 | 65.9 |
| **Swin UNETR** | **public → vertebrae** | **71.3** | **91.6** | **89.0** | **76.0** |
| 3D MAE | - | 35.1 | 58.8 | 70.3 | 30.2 |
| **3D MAE** | **vertebrae** | **75.2** | **92.9** | **92.5** | **82.0** |
| Video MAE | - | 33.3 | 84.5 | 65.5 | 23.7 |
| Video MAE | 650.000 video clips[6] (public) | 80.1 | 93.9 | 96.5 | 89.1 |
| **Video MAE** | **public → vertebrae** | **84.1** | **95.3** | **96.8** | **90.4** |
| Video-CT MAE | - | 34.8 | 68.1 | 67.6 | 28.3 |
| Video-CT MAE | 650.000 video clips (public) | 85.7 | 95.5 | 98.0 | 92.5 |
| **Video-CT MAE (ours)** | **public → vertebrae** | **88.4** | **96.4** | **98.2** | **93.2** |

Table 3: Comparison with other self-supervised pretraining methods

Models leveraging both public and task-specific pretraining consistently surpassed those limited to public data pretraining or no pretraining at all. This was particularly evident in transformer-based models (ViT UNETR, Swin UNETR, MAE, and our Video-CT MAE), underscoring the critical role of pretraining in these architectures. Models Genesis demonstrated strong performance without pretraining, suggesting that CNN-based models may be less reliant on extensive pretraining for smaller datasets. Our Video-CT MAE method proved to be the most effective, surpassing all other evaluated methods across all metrics.

### 5.3. Vertebra Fracture Detection

We evaluate our method for vertebral fracture detection by comparing it with an existing technique and common 3D classification architectures in medical settings. In addition, we show the challenges of training 3D ViTs from scratch, which inspired our proposed approach.

| Method | F1 (%) | ACC (%) | AUC (%) | AP (%) |
|---|---|---|---|---|
| DenseNet121 (Huang et al., 2017) | 73.7 | 92.1 | 92.1 | 81.1 |
| DenseNet169 (Huang et al., 2017) | 72.9 | 91.7 | 93.3 | 83.9 |
| ResNet18 (He et al., 2016) | 81.8 | 94.6 | 95.4 | 89.6 |
| ResNet50 (He et al., 2016) | 79.5 | 93.7 | 94.6 | 85.0 |
| ViT-B (Dosovitskiy et al., 2020) | 32.2 | 61.2 | 63.0 | 19.7 |
| ViT-L (Dosovitskiy et al., 2020) | 33.5 | 55.0 | 64.7 | 22.0 |
| Engstler et al. (2022) | 85.1 | 95.4 | 96.2 | 89.1 |
| **Video-CT MAE (ours)** | **88.4** | **96.4** | **98.2** | **93.2** |

Table 4: Comparison with state-of-the-art vertebra fracture detection

Our approach significantly outperforms conventional classification architectures and shows superior results to the vertebral fracture classification method of Engstler et al. (2022). Our presented method successfully implements 3D ViTs in a challenging 3D medical context, characterized by the scarce and imbalanced labeled data.

## 6. Conclusion

Our novel approach adapts video-based transfer learning through task-specific self-supervised domain adaptation, successfully enabling Vision Transformers to address vertebral fracture detection in 3D CT scans. This study not only advances the state-of-the-art in vertebral fracture detection but also showcases the potential of task-specific pretraining for other medical image analysis tasks. Future research could explore the creation of task-specific pretraining datasets for various applications and evaluate the generalizability of our approach. By addressing the challenge of limited data in medical image analysis, our work offers a promising solution for improving patient care through accurate, interpretable, yet resource-efficient methods for advancing clinical decision-making support.

---

3. Models Genesis 3D pretraining dataset: LUNA16 (Setio et al., 2017)

4. ViT UNETR pretraining dataset: TCIA-Covid19 (An et al., 2020)

5. Swin UNETR pretraining datasets: TCIA-Covid19 (An et al., 2020), LUNA16 (Setio et al., 2017), HNSCC (Grossberg et al., 2020), LiDC (Armato III et al., 2011), TCIA Colon (Johnson et al., 2008)

6. Video MAE pretraining dataset: Kinetics-700 (Carreira et al., 2019)

## Acknowledgments

The authors acknowledge the financial support by the Federal Ministry of Education and Research of Germany (BMBF) under project DIVA (FKZ 13GW0469C). This work was partially funded via the EVUK programme ("Next-generation AI for Integrated Diagnostics") of the Free State of Bavaria.

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

# Appendix A. Experimental Setup

## A.1. Hardware Specifications

All trainings conducted in this paper were performed using a single Nvidia A40 GPU.

## A.2. Preprocessing Pipeline

Figure 2 presents an overview of the preprocessing pipeline, while Figure 3 provides a detailed view of a single vertebra crop produced by the pipeline.

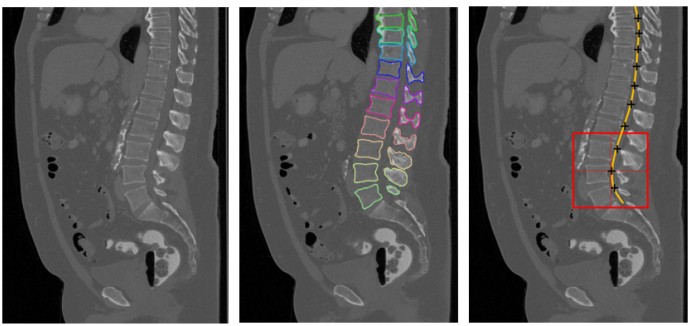

Figure 2: Overview Preprocessing Pipeline: 1) input CT scan 2) segmentation mask output of segmentation model 3) cropping of $96 \times 96 \times 96$ vertebra crop based on spline running through spine (orange)

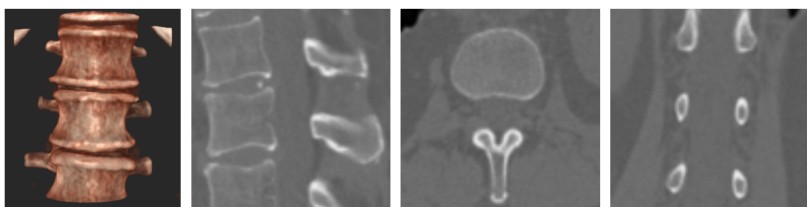

Figure 3: Visualization of Centered $96 \times 96 \times 96$ Vertebra Crop: 1) 3D visualization 2) mid-sagittal slice 3) mid-axial slice 4) mid-coronal slice

## A.3. Unlabeled Vertebra Pretraining Dataset

This appendix lists additional references related to the public datasets used in our study. These works, while not cited directly in the main text, provide important background and context for the datasets. Their inclusion here acknowledges their contribution to the field and offers readers further resources on the topic. (Clark et al., 2013) (Johnson et al., 2008) (Bejarano et al., 2019) (Liebl et al., 2021) (Löffler et al., 2020)

### A.4. Labeled Vertebra Downstream Task Dataset

Table 5 provides a detailed summary of the labeled dataset from Klinikum Rechts der Isar (Munich) (Foreman et al., 2024). This dataset was used for the downstream task trainings.

| | classes | |
|---|---|---|
| **split** | **no fracture** | **fracture** |
| training | 3,336 | 556 |
| validation | 947 | 211 |
| test | 1,022 | 173 |
| **TOTAL** | **5,305** | **940** |

Table 5: Labeled Vertebra Downstream Task Dataset

### A.5. Implementation Details

**Self-supervised Domain Adaptation Pretraining** In our experimental setup, we employ the video MAE pretraining method by Feichtenhofer et al. (2022) for both video and domain-specific vertebra CT pretraining. For the MAE model we use the ViT-Large version (Dosovitskiy et al., 2020) as encoder and a decoder depth of 4. We use convolutional patch embeddings of size $2 \times 16 \times 16$. A masking ratio of 0.8 is applied. We initialize domain-specific vertebra CT pretraining with weights from Kinetics-700 pretraining, utilizing positional encoding cropping. Training consists of 100 epochs with a batch size of 8, using the AdamW optimizer and a cosine learning rate scheduler (base learning rate 1e-3). Data preprocessing involves clipping HU values to the range -1000 to 1000 and then scaling them to 0-1. To match the RGB video format, we replicate the HU values. The input volumes have a shape of $16 \times 96 \times 96$, following the described sampling technique.

**Downstream Task Finetuning** The MAE pretraining architecture is adapted for the subsequent classification task, exclusively utilizing the ViT-Large encoder. Global average pooling is applied to the output tokens, followed by a linear classification layer for binary classification. Finetuning is done for 50 epochs, employing a batch size of 64, Adam optimizer, and a cosine learning rate scheduler with warmup (base learning rate 1e-5). Class weighting is introduced to the cross entropy loss to handle data class imbalance. Data preprocessing remains consistent with pretraining. The labeled vertebra dataset is split into training, validation, and testing sets at a 60%/20%/20% ratio, ensuring patient-level separation to prevent test-leakage.

## Appendix B. Results and Discussion

### B.1. Ablation Study - Positional Encoding Cropping

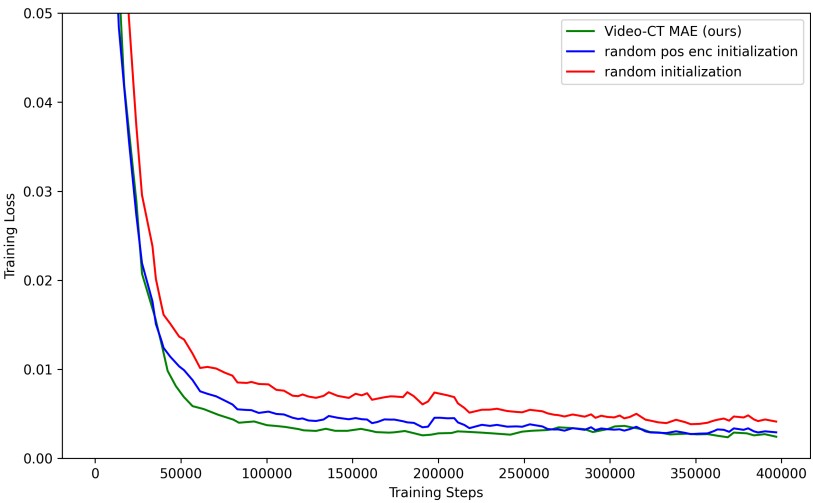

Figure 4: Ablation Study - Positional Endocing Cropping: Pretraining Loss Comparison

### B.2. Method Comparison - Implementation Details Baseline Methods

This appendix details the implementation of the pretraining methods in our study, adapted to our dataset setup. Following our Video-CT MAE method's dataset setup (see Experimental Setup 4), we first pretrain on the unlabeled vertebra dataset, then finetune on the labeled dataset, using $96 \times 96 \times 96$ vertebra crops.

### B.2.1. MODELS GENESIS

**Pretraining (Zhou et al., 2021)**   To better align with our vertebra crops, we increased the input size from the original $64 \times 64 \times 32$ to $96 \times 96 \times 96$, while maintaining all other original settings. Pretraining was initialized with publicly available pretrained weights.[7]

**Finetuning**   We modified the pretrained model by utilizing solely its encoder and appending a binary classification head consisting of two layers.

### B.2.2. VIT UNETR

**Pretraining**   We followed the official public implementation and initialized pretraining with publicly available pretrained weights.[8]

**Finetuning**   We used the pretrained ViT-Base encoder for the downstream task by appending a classification token and integrating a classification head on top of this token.

---

7. https://github.com/MrGiovanni/ModelsGenesis/tree/master
8. https://github.com/Project-MONAI/tutorials/tree/main/self_supervised_pretraining

### B.2.3. Swin UNETR

**Pretraining ([Tang et al., 2022](#))** We followed the official implementation and initialized pretraining with publicly available weights. The Swin encoder was initialized with public weights, but the decoder was randomly initialized due to a shape mismatch.[9]

**Finetuning** We exclusively utilized the Swin ViT encoder. Global average pooling was applied to the output tokens, followed by a binary classification head.

### B.2.4. MAE

**Pretraining ([He et al., 2022](#))** We adapted this method for our 3D CT data with minimal changes, such as transitioning from 2D to 3D convolutions in the patch embedding layer and modifying positional encoding to 3D. Apart from these adaptions, we maintained the original settings, choosing ViT-Base encoder and a masking ratio of 0.75.[10]

**Finetuning** In the finetuning phase, we exclusively utilized the ViT-Base encoder, complementing it with a binary classification head attached to the classification token.

---

9. https://github.com/Project-MONAI/research-contributions/tree/main/SwinUNETR/Pretrain
10. https://github.com/facebookresearch/mae

