# OpenReview forum: "Video-CT MAE: Self-supervised Video-CT Domain Adaptation for Vertebral Fracture Diagnosis"
_MIDL.io/2024/Conference — MIDL 2024 Poster_

### Official Review · Reviewer_uWo6 · 2024-02-28

**Confidence:** 5
**Preliminary Rating:** 3
**Recommendation:** Poster
**Final Rating:** 3.5

**Summary:**

Authors propose an approach to solve the task of binary classifying fractures on CT images of vertebra. The approach is based on three stages, (1) transfer learning from video ViT model (2) pretraining with MAE on a large unlabeled dataset (3) fine-tuning on the desired task with labeled data. The authors also use cropped positional embeddings and slice sampling to adapt the new task input size to the pretrained model. Ablation experiments are provided regarding the different stages and proposed modules of the method, as well as comparisons to other relevant transfer learning and pretraining approaches.

**Strengths:**

- Overall, the paper is pleasant to read (well-written, and organization is clear);
- Clear understanding of and relevant medical application tackled;
- The methodology is well-presented, the evaluation experiments and ablations are appropriate;
- Reproducibility seems correct (but code is not available at the time of reviewing).

**Weaknesses:**

- either lack of novelty (I deem it is quite common to use (1) transfer learning from natural images or video models (2) pretraining with some self-supervised method such as MAE on large unlabeled data (3) fine-tuning on the desired task with labeled data stages) or lack of evidence that such approach could be more broadly used for medical datasets (only one task solved); Especially regarding the fact that authors claim to work under small data regime but they successfully curated 27k samples.
- lack of comparison methods regarding studies solving the same task (only comparison methods to similar methodologies, not initially proposed to solve medical-related tasks), comparison expected at least from the state-of-the-art presented in the related work;

**Detailed Comments:**

Major comments:
- not sure Domain Adaption is a correct keyword here when having to pretrain on large unlabeled data + fine-tuning the whole model after on labeled data, while testing from data from the same dataset;
- The two first contributions need to be more specific, using transfer learning is not new (even from video pretrained models), the benefits of using self-supervised methods on top of transfer learning before solving a task is well-known, the used slice sampling is like a sliding window solution (plus see comment giving more details on this slice sampling part), as well as the cropped positional embedding solution to different input sizis [A]
- third contribution is the large-scale dataset, how more precisely? the dataset will be made public? Will the code to easily collect it be available?
- section 3: "Initializing CT models with ImageNet weights appears suboptimal" does it correspond to no video pretraining ablation, or this ablation is pretrained from scratch? I would like to see if even with the SSL stage, the Video pertaining is better than ImageNet pretraining
- section Slice Sampling: detail how the sampling is performed: randomly, uniformly? would a sliding window even be better? how does it affect computational burden compared to training a model on the 96x96x96 size, e.g. by interpolating (or extrapolating) the positional embeddings instead of cropping?
- does the segmentation model and code for cropping provided by the ImFusion GmbH are related to a research article or open code? if yes please add references, if not maybe briefly explain the approach.
- the datasets includes 3446 CT volumes, is it from different patients only?

[A] Kim et al., Region-Aware Pretraining for Open-Vocabulary Object Detection with Vision Transformers, CVPR, 2023.

Minor comments:
- Table 1), 2): maybe highlight also 2nd best performing method (with underlined or italic0 for clarity?
- Figure 1: room to also detail there the slice sampling and positional embeddings cropping strategies used?

**Justification Of Final Rating:**

During the rebuttal period, authors provided a supplementary relevant comparison method on vertebra classification, as well as a few clarifications on the introduction and results analysis.
However, as mentioned by all reviewers, this manuscript provides an interesting message on pretraining and self-supervised learning (on a specific dataset), but overall the technical novelty and study impact remain limited.
So I would keep my rating to borderline, on the accept side.

**Justification Of The Preliminary Rating:**

The paper is clear, and well-written; overall, the methodology and evaluation are appropriate. But accounting for the rather weak methodological contributions, I would expect more in-depth analysis and demonstration of the proposed method's generalizability.

**Questions To Address In The Rebuttal:**

- Comparison to at least the SOTA method to solve this specific task;
- Clarify, specify contributions;
- Add at least one dataset to demonstrate the relevance of the proposed method to other medical applications (and, to me, accounting for the rather weak methodological novelty);
- Answer questions in the major comments.

**Special Issue:**

No

---

> ### Author Response · Authors · 2024-03-18
> **Revised submission: additional experiments; vertebral fracture detection SOTA comparison; more precise contributions and motivation of proposed method**
>
> ## Comment:
> Thank you for your comprehensive review and the time you dedicated to understanding our work. Your insightful comments and constructive suggestions are greatly appreciated, as they have helped us refine and improve the quality of our paper. Based on your feedback, we have made substantial revisions to the manuscript, with a particular focus on enhancing the experimental design and improving the overall clarity of the paper. We believe these changes have strengthened the work and made it more impactful.
>
> ## Questions to address:
> We appreciate the reviewer's emphasis on the necessity of benchmarking our method against established techniques in vertebral fracture detection to position our work in the community. To address this, the revised version of our paper includes a comprehensive comparison table. Table 4 contrasts our approach with a state-of-the-art method in vertebral fracture detection and commonly used 3D classification architectures within the medical field. Furthermore, we have incorporated details of our preliminary experiments using ViTs, highlighting the challenges and our motivation behind the proposed method for implementing ViTs (with their inherent interpretability ability) in our challenging task.
>
> We would like to thank the reviewer for bringing our attention to the initially unclear definition of the paper's contributions. In the revised version, we have improved both the abstract and the introduction to more clearly and precisely articulate the paper's motivation and key contributions. We highlight that our approach, which uniquely combines existing methods, represents a novel contribution. Specifically, this is the first method, to our knowledge, applying transfer learning from the video domain to initialize task-specific, self-supervised pretraining and not directly initialize the downstream task. This approach enables 3D ViT-based models to effectively handle complex 3D medical data in scenarios with limited labeled information.
>
> The reviewer's suggestion to evaluate our approach's generalizability across different medical applications is interesting. Our current focus is on underscoring the significance of task-specific pretraining. Regrettably, we could not apply our method to a different medical application with a similar data setup, allowing task-specific pretraining. Nevertheless, our paper lays the groundwork for future research, potentially paving the way for developing a generic pretrained model for full-body CT scans.
>
> ## Other concerns:
> Addressing the initial absence of code during the first review phase, we have now published it. The code is accessible through the link provided in the paper's abstract.
>
> We recognize the validity of the reviewer's suggestion to publish our dataset. Unfortunately, licensing restrictions on the original datasets prevent us from releasing the vertebrae crops. However, we can make the code used to create the dataset available. Other researchers can replicate the dataset using this code, together with the vertebral segmentation model from ImFusion GmbH. Detailed information on the preprocessing pipeline is included in the appendix of our paper. We also added a citation to a relevant paper by ImFusion GmbH on vertebra localization.
>
> In the comparison of ImageNet and video weights, our downstream task analysis revealed that ImageNet weights did not surpass random initialization in our data setup. Consequently, this insight led us to favor video data pretrained weights, demonstrating superior performance.
>
> To clarify our dataset setup: We curated 27K unlabeled vertebrae crops with no fracture label from public datasets originating from 3,500 different patients. We have 6K labeled vertebra crops, including 940 fractured ones, from 457 distinct patients. We think it is valid to say that we work under a small data regime.
>
> We want to thank the reviewer for highlighting the reference on positional encoding cropping, of which we were unaware. We've now included this citation in our paper.
>
> We've updated the paper to clarify our sampling scheme. In the training phase, we sample 16 uniformly distributed sagittal slices. Each training step involves applying a consistent random shift to the chosen slice indices. During the evaluation phase, we maintain our initial sampling strategy but do not shift the selected slice indices. Instead, we use all six possible 16x96x96 samples covering the entire vertebral volume for prediction.
> Our sampling method, we believe, is more effective than sliding window as it ensures the entire volume is represented in each sample, whereas sliding window may result in only a few windows containing the fracture.
> We haven't explored 96x96x96 training due to necessary additional changes in the patch embedding layer. Future work may consider positional encoding inter/extrapolation in combination with weight inflation in the patch embedding layer to make 96x96x96 with 16x16x16 input patches possible.

---

> > ### Comment · Reviewer_uWo6 · 2024-03-22
> > **Response to rebbutal**
> >
> > Thank you for your detailed answer and modifications to the manuscript.

---

### Official Review · Reviewer_TB9b · 2024-02-29

**Confidence:** 4
**Preliminary Rating:** 3
**Final Rating:** 4

**Summary:**

This paper introduces improving vertebral fracture diagnosis through self-supervised learning, utilizing a unique video-CT domain adaptation technique. Comparative analysis reveals that the proposed approach outperforms existing self-supervised learning methods in downstream vertebra classification tasks.

**Strengths:**

- The structure of the manuscript is well-organized with figures and tables effectively contributing to the comprehensibility of the content.
- The proposed method is promising and meaningful, applying video data to tackle challenges within the domain of medical imaging.
- The article is thorough in providing ample details that facilitate a full understanding of the employed technique.

**Weaknesses:**

- The provided code link contains no actual code.
- Ablation experiments should focus on the impact of Video Pretraining on methods like Models Genesis (Zhou et al., 2021), ViT UNETR (Tang et al., 2022), Swin UNETR (Tang et al., 2022), and MAE (He et al., 2022); rather than being trained on the unlabeled vertebra dataset, which is evidently a useful strategy.
- Regarding Table 2, the authors are encouraged to provide experimental results for additional combinations of Ablation Elements.
- The approach of video MAE pretraining followed by positional encoding cropping, vertebra CT pretraining, and downstream task fine-tuning seems to be quite straightforward, with limited technical innovation.

**Detailed Comments:**

In Figure 4, random initiation shows comparable performance to the proposed Video-CT MAE, please explain the reason.

**Justification Of Final Rating:**

The rebuttal has addressed most of my concerns, including novelty, the initial absence of code during the first review phase, and additional ablation experiments.
I have revised my score to Weak Accept.

**Justification Of The Preliminary Rating:**

I acknowledge that the proposed method is promising and meaningful but worried about limited novelty and experiments.

- Ablation experiments should focus on the impact of Video Pretraining on methods like Models Genesis (Zhou et al., 2021), ViT UNETR (Tang et al., 2022), Swin UNETR (Tang et al., 2022), and MAE (He et al., 2022); rather than being trained on the unlabeled vertebra dataset, which is evidently a useful strategy.
- The approach of video MAE pretraining followed by positional encoding cropping, vertebra CT pretraining, and downstream task fine-tuning seems to be quite straightforward, with limited technical innovation.

**Questions To Address In The Rebuttal:**

Concerns about limited novelty and experiments.

---

> ### Author Response · Authors · 2024-03-18
> **Revised submission: additional experiments; two new ablation experiments; more precise contributions and motivation of the proposed method**
>
> ## Comment:
> We greatly appreciate the feedback and valuable recommendations provided by the reviewer. We aim to ensure that our paper is a useful and accessible resource for the entire MIDL community and provides a strong foundation for future studies. In response to the comments received, significant improvements have been made, with particular attention paid to refining the experiments for ablation studies and task-specific pretraining. We additionally added experiments to position our work in the community of vertebral fracture detection.
>
> ## Questions to address:
> We appreciate the reviewer's emphasis on the need for comprehensive experiments. In response, we have enriched our study with additional experiments, particularly focusing on the impact of task-specific pretraining. Our expanded self-supervised pretraining methods comparison now clearly illustrates that task-specific dataset pretraining enhances performance compared to using public weights or training from scratch.
>
> To further illustrate the impact of our research on vertebral fracture detection, the revised paper includes new experiments. In Table 4, we introduce a comparison that contrasts our method with a state-of-the-art approach for vertebral fracture detection and other established 3D classification architectures for medical data. Furthermore, we have incorporated details of our preliminary experiments using ViTs, highlighting the challenges and our motivation behind the proposed method for implementing ViTs. These additions underscore our method's significant advancements over traditional classification architectures and its superior performance compared to a state-of-the-art model in vertebra fracture classification. Our analysis of the ViT models demonstrates that we successfully implemented a method that makes 3D ViTs effective in the challenging 3D medical context, overcoming the hurdles of limited and imbalanced labeled data.
>
> We appreciate the reviewer's perspective on the perceived novelty of our method. We would like to highlight that our approach, which uniquely combines existing methods, represents a novel contribution. Specifically, this is the first method, to our knowledge, applying transfer learning from the video domain to initialize task-specific, self-supervised pretraining. This approach enables 3D ViT-based models to effectively handle complex 3D medical data in scenarios with limited labeled information. We believe our method lays a significant groundwork for future explorations in this area.
>
> ## Other concerns:
> Addressing the initial absence of code during the first review phase, we have now published it. The code is accessible through the link provided in the paper's abstract.
>
> We find the suggestion of pretraining other methods on video datasets interesting. However, the aim of our work is to demonstrate the adaptation of publicly available weights to task-specific datasets. This approach allows effective pretraining with limited computational resources, bypassing the extensive time and hardware requirements of pretraining on large datasets such as Kinetics-700. Indeed, the proposed idea holds merit and presents a valuable direction for future research.
>
> In response to the suggestion for additional ablation experiments, we have included two new ablation experiments. The first experiment (Experiment 5) compares the performance of domain-specific pretraining and downstream task finetuning using the original 16x224x224 video format versus the adjusted 16x96x96 format. Although the results are similar, we observed a significant increase in training time with the original video format. The second experiment highlights the crucial role of positional encoding cropping when directly transitioning from video pretraining to the downstream task, bypassing the task-specific pretraining stage.
>
> Addressing the comment on the comparable performance depicted in the loss curves between random positional encoding initialization and positional encoding cropping, we have refined our discussion in the text. We argue that despite the similar performance, positional encoding cropping offers a distinct advantage by facilitating faster convergence. This efficiency leads to a reduction in the number of training epochs required, consequently saving valuable training time. The reason for this is that the positional encodings can be relearned during the self-supervised training. When not relearning the positional encodings using the self-supervised domain adaptation step, the results on the downstream task get worse (see ablation experiment 6).

---

### Official Review · Reviewer_Ezbg · 2024-03-01

**Confidence:** 5
**Preliminary Rating:** 2
**Final Rating:** 3.5

**Summary:**

Authors propose to use self-supervised fine-tuning of a
transformer-based architecture that was pre-trained using videos,
before fine-tuning for task. They created a large data sets of
volumetric CT crops over vertebrae and fine-tuned a ViT-Large using
the same strategy as the original pre-training with videos, but using
vertebrae crops. Some adaptations were necessary. This self-supervised
transfer of the original ViT proved to be useful in ablation studies
and comparisons with other architectures.

**Strengths:**

- The final message I get from the article is
  important. Self-supervised fine-tuning for transferring pre-trained
  models for a given task can improve performance.
- Ablation studies are well designed. They are pretty useful for the
  main message.
- Combining publicly available data sets and smaller labeled data sets
  is a good strategy for the self-supervised transfer.

**Weaknesses:**

- The introduction and the abstract can be sharpened to more clearly
  state the main contribution of the work. Especially, the abstract is
  a bit cryptic. I believe sharpening the text would greatly
  facilitate the dissemination of the work. Effectively, the paper
  proposes to use self-supervised transfer learning for a specific task.
- Technical novelty of this work remains on the lower end.
- The transfer self-supervised learning is quite task
  specific. Authors perform this training on volumetric CT-crops
  around vertebrae. It is unclear, how much this can be extended to
  other CT-related tasks.
- While the ablation studies are great, the comparisons with existing
  techniques for the same task is not very informative. To the best of
  my understanding, authors do not compare with previous work on
  fracture detection. Since the paper is largely an empirical study, I
  strongly believe a proper comparison is necessary. This would allow
  readers to compare the pre-training strategies with existing models
  for fracture detection. I believe comparisons are simply different
  generic architectures.

**Detailed Comments:**

Please check the weaknesses I mention above for the detailed comments.

**Justification Of Final Rating:**

I am happy with the revisions authors made to the article. This improves the quality of the article by a large margin.
However, it also means they applied a lot of change to the article. If this is acceptable, I recommend accepting the article. If this is an issue, I believe the article should not be accepted in this cycle and authors should be encouraged to submit to the following conference.

**Justification Of The Preliminary Rating:**

- Technical novelty is on the lower end.
- Experimental design does not compare with prior work focusing on
  fracture detection. Hence, it becomes hard to position the work in
  the community.
- While the rebuttal can answer some of the weaknesses, I am afraid
  it may also change the paper too much.

**Questions To Address In The Rebuttal:**

- For the rebuttal, it would be important to do comparisons with
  existing techniques for fracture detection. Otherwise, it is not
  possible to understand the value of the proposed technique.
- It would also be interesting to demonstrate performance in the case
  where the self-supervised pre-training is not restricted to
  vertebrae crops but performed on the entire CT volumes. This would
  explain whether the methodology can create pre-trained networks that
  can be used for other CT-related tasks.

---

> ### Author Response · Authors · 2024-03-18
> **Revised submission: additional experiments; vertebral fracture detection SOTA comparison; more precise contributions**
>
> ## Comment:
> We sincerely thank the reviewer for the insightful comments and constructive suggestions regarding our paper. We want the paper to be accessible and useful for the entire MIDL community as we think the paper sets a good foundation for future research. In response to the feedback received, we have undertaken substantial revisions, particularly focusing on enhancing the experimental design of our paper.
>
> ## Questions to address:
> We appreciate the reviewer's emphasis on the necessity of benchmarking our method against established techniques in vertebral fracture detection to position our work in the community. To address this, the revised version of our paper includes a comprehensive comparison. The results added in Table 4 contrast our approach with both a state-of-the-art method in vertebral fracture detection and commonly used 3D classification architectures within the medical field. Furthermore, we have incorporated details of our preliminary experiments using ViTs, highlighting the challenges and our motivation behind the proposed method for training ViTs on challenging 3D tasks with limited data. These additions underscore our method's improvement over established classification architectures and its superior performance compared to the state-of-the-art in vertebra fracture classification.
>
> The reviewer's suggestion about evaluating generalizability through pretraining on the entire CT volume (not only vertebra crops) is interesting, and we compare it to Model Genesis pretrained weights that followed this approach. However, within the scope of this paper, we focused on improving a challenging downstream task as opposed to a general feature extractor for CT images. We show experimentally that this task-specific pretraining with unlabelled data is highly effective in improving the downstream task performance as highlighted in the now extended Table 3. We agree that for future research it would be interesting to investigate the generalizability of this approach to other medical applications.
>
> ## Other concerns:
> Thank you for highlighting the need for improved clarity in our abstract and introduction. In response, we have made extensive revisions in both sections of the revised version of our submission. These improvements aim to clarify the motivation behind our work and to explicitly outline the main contributions.
>
> Furthermore, we have expanded our experimental section to more comprehensively demonstrate the impact of task-specific pretraining. These additional experiments robustly illustrate that pretraining on a task-specific dataset notably enhances performance compared to utilizing publicly available weights or training from scratch.
>
> We appreciate the reviewer's perspective on the perceived novelty of our method. We would like to highlight that our approach, which uniquely combines existing methods, represents a novel contribution. Specifically, this is the first method, to our knowledge, applying transfer learning from the video domain to initialize task-specific self-supervised pretraining. This approach enables 3D ViT-based models to effectively handle complex 3D medical data in scenarios with limited labeled information. This contribution is evident in the gain in performance over existing self-supervised pretraining methods for medical 3D data and naive transfer learning from video weights shown in Table 3. We believe our method lays a significant groundwork for future explorations in this area.

---

### Decision · Program_Chairs · 2024-04-05

Accept (Poster)